# Stride-TCN for Energy Consumption Forecasting and Its Optimization

Le Hoang Anh [1,†], Gwang Hyun Yu [1,†], Dang Thanh Vu [1], Jin Sul Kim [1], Jung Il Lee [2], Jun Churl Yoon [3,*] and Jin Young Kim [1,*]

1 Department of ICT Convergence System Engineering, Chonnam National University, 77, Yongbong-ro, Buk-gu, Gwangju 61186, Korea
2 Korea Electric Power Research Institute (KEPRI), 105, Munji-ro, Yuseong-ku, Daejeon 34056, Korea
3 Korea Electric Power Corporation (KEPCO), 55, Jeollyeok-ro, Jeollanam-do, Naju-si 58322, Korea
* Correspondence: hiyoon@kepco.co.kr (J.C.Y.); beyondi@jnu.ac.kr (J.Y.K.)
† These authors contributed equally to this work.

**Abstract:** Forecasting, commonly used in econometrics, meteorology, or energy consumption prediction, is the field of study that deals with time series data to predict future trends. Former studies have revealed that both traditional statistical models and recent deep learning-based approaches have achieved good performance in forecasting. In particular, temporal convolutional networks (TCNs) have proved their effectiveness in several time series benchmarks. However, presented TCN models are too heavy to deploy on resource-constrained systems, such as edge devices. As a resolution, this study proposes a stride–dilation mechanism for TCN that favors a lightweight model yet still achieves on-pair accuracy with the heavy counterparts. We also present the Chonnam National University (CNU) Electric Power Consumption dataset, the dataset of energy consumption measured at CNU by smart meters every hour. The experimental results indicate that our best model reduces the mean squared error by 32.7%, whereas the model size is only 1.6% compared to the baseline TCN.

**Keywords:** temporal convolutional networks; deep learning; time-series forecasting

## 1. Introduction

Research on energy usage management has a long tradition, especially about energy consumption. According to Enerdata (https://yearbook.enerdata.net/electricity/electricity-domestic-consumption-data.html, accessed on 18 July 2022) record in 2021, electricity accounts for 10% of all types of global energy, mostly used by China (42%), the United States (21%), and India (7%). Forecasting models are necessary to optimize an energy consumption program to reduce energy loss efficiently. Many forecasting models, both classical statistical methods and deep learning, are available.

The time-series forecasting problem is usually overcome by classical statistical methods, such as autoregressive (AR) [1] models and the Gaussian process [2], or methods using deep learning, such as LSTM, GRU, and Transformers [3]. Although the above methods operate well in specific circumstances [4], their potential in practice is limited due to heavy computation.

In this paper, we propose the stride–dilated temporal convolutional networks (TCNs), a family of lightweight models for predicting energy consumption. The proposed method is based on the periodic patterns of time series that are usually visible in energy consumption data. Given the capability of detecting periodic patterns, the proposed model is capable of automatically focusing on learning the important parts of data to make predictions. To predict a period in the future, we focus on extracting information from moments in the past. We hypothesize that there are only a few important time points when forecasting, and most of the history has a low correlation; therefore, we can ignore it to reduce the computational burden. Moreover, we observe that different time series data have cyclic patterns that differ

from others. Therefore, we suggest using a search algorithm to determine an optimized TCN architecture with appropriate stride hyperparameters; thus, this paper adopted Bayes optimization.

The aim of this work is two-fold:

- We propose a new TCN architecture with performance on par with state-of-the-art models on several benchmarks, but the number of parameters is greatly reduced based on the stride mechanism. We search for the best stride hyperparameter, representing cyclic patterns on the data.
- We introduce a dataset of electrical energy consumption measured at Chonnam National University (CNU), South Korea. Along with the dataset, we also present the baseline benchmark.

The rest of the paper is organized as follows: Section 2 presents the background work. Section 3 details the methodology, and Section 4 describes the experiments on the dataset. Then, Section 5 discusses the work and presents a conclusion. Additionally, we publish the source code and datasets used for the experiments (https://github.com/andrewlee1807/tcns-with-nas).

## 2. Background

The staleness of the model, quality of data, and long prediction range are usually ill-posed problems in the case of time-series forecasting [5]. Various methods have been proposed to solve the above problems, not only with classical statistics but also with recent deep-learning-based techniques.

As a representative of traditional methods, autoregressive integrated moving average (ARIMA) [5] has been used for many years in the field of time series modeling. An ARIMA model combines AR and the moving average to account for seasonality, long-term trends, autoregression, and autocorrelation embedded in the data. However, long-term models are inevitably prone to overfitting and high computational costs [3].

Moreover, deep learning techniques, such as CNNs and RNNs, are also introduced and widely applied for time series forecasting. The advantage that can be gained using a deep learning approach for time series forecasting is that it does not require manually dealing with in-depth data beforehand because representative features are automatically extracted through a training process. The various extensions of RNNs, such as long short-term memory (LSTM) and gated recurrent units (GRU), are suited for time series forecasting [3]. However, these memory and gate-based models still suffer the problem of a long-time dependency on time-series data. In particular, the number of parameters significantly increases even when only a short additional time interval is considered [6].

The TCN [7] is proposed to consider the long-time dependency issue in time series data. As far as we know, the TCN is currently a prominent model in time series forecasting due to two significant modifications: dilation and causal one-dimensional (1D) convolutional layers. The causal 1D convolutional layer offers a learnable kernel filter with proper padding only on the past side to avoid overprediction. Moreover, dilation is the main factor that grants the TCN the ability to enlarge the history coverage via stacking layers. Therefore, the TCN is very effective in dealing with time-series problems and is easily extensible. However, to ensure the range of history for forecasting, a TCN often stacks many convolutional layers on top of each other, significantly increasing the number of parameters and requiring a considerable time to train the model. With the principles of TCN in mind, we develop a simplified architecture that uses correlation factors from a dataset to achieve a compact model that is robust in terms of model lower complexity and robust performance.

A large number of existing studies in the broader literature have examined TCN and its variants for various tasks and types of datasets. Gan et al. employed TCN with an interval width adjustment strategy for wind speed forecasting [8]. For the same task of wind speed forecasting, Li et al. proposed a framework that combined patch transformation, mode decomposition with adaptive noise, and TCN [9]. In mechanical systems, Cao

et al. introduced TCN with a residual self-attention mechanism for remaining useful life prediction [10]. For video segmentation, Dipika et al. proposed coarse to fine multi-resolution encoder–decoder TCN to ensure smoothness and temporal coherency [11]. Ma et al. presented densely connected TCN with a squeeze-and-excitation block and attention mechanism to increase the receptive field's size for solving the lip-reading problem in videos [12].

## 3. Methodology

In this section, the time series forecasting problem is formulated first. In addition, the TCN model is used as the method in the comparative evaluation. Finally, the stride-TCN is introduced.

### 3.1. Time Series Forecasting Problem

First, we highlight the nature of the time-series forecasting task. Given an input sequence $X = \{x_0, \ldots, x_T\}$, where the task is to predict the outputs $Y = (\hat{y_0}, \ldots, \hat{y_P})$ each time, a key constraint is that, from an ordered number of $T$ observed data points, $P$ data points must be immediately predicted chronologically. Formally, a modeling network is any function $f : X \rightarrow Y$ that produces the mapping

$$Y = f(X) \tag{1}$$

Building function $f$ is the process of learning to find optimal parameters of network $f$ from a set of time series $\left\{ x_{0:T}^{(i)} \right\}_{i=1}^{N}$ that denotes the future time series as $\left\{ x_{(T+1):(T+P)}^{(i)} \right\}_{i=1}^{N}$, where $N$ is the number of series, $T$ represents the length of the historical observations, and $P$ indicates the length of the forecasting horizon. The learning process is the process of minimizing the error function $L(Y, f(X))$ between the actual output and predictions. Time-series forecasting generally focuses on the prediction of real values, usually loss functions, such as the mean squared error (MSE), mean absolute error (MAE), or its variants (mean absolute percentage error, root mean square error, etc.). The MSE is greater for learning the outliers in the dataset, whereas MAE is good for ignoring outliers. However, in some cases, the data are less sensitive to outliers, and those points should not have high priority. Therefore, the Huber loss combines the proposed MAE and MSE and solves this problem [6]. The mathematical form of the Huber loss is:

$$L(Y, f(X)) = \begin{cases} \frac{1}{2}(Y - f(X))^2 & \text{for } |Y - f(X)| \leq \delta \\ \delta |Y - f(X)| - \frac{1}{2}\delta^2 & \text{otherwise} \end{cases} \tag{2}$$

where the $\delta$ parameter is sensitive to outliers. In this study, we used the Huber loss as the error function to calculate the difference between the current and expected output where the $\delta$ parameter is fine-tuned by the algorithm.

### 3.2. TCN Architecture

A reasonable volume of training data was available in this work; thus, we decided to use deep learning models to predict hour-ahead energy consumption. The TCN is a variation of the CNN for sequence modeling tasks by combining aspects of the RNN and CNN architectures. The TCN achieves better performance than RNNs in many tasks while avoiding the common drawbacks of recurrent models, such as the exploding/vanishing gradient problem or lack of memory retention [7]. The original architecture of the TCN (Figure 1) introduced includes two components: dilated and causal 1D convolutional layers [13], which smooth the input time series. Thus, we do not need to add the rolling mean or rolling standard deviation values in the input features.

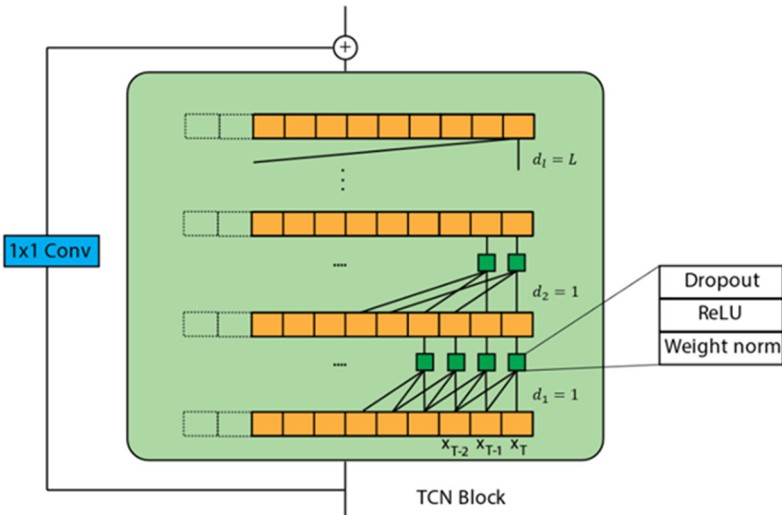

**Figure 1.** TCN architecture.

Dilated convolution can be applied in the long information dependency problem of the sequence to determine the output $\mathcal{Y}$ at position $t$ for a sequence input $x \in \mathbb{R}^n$, expressed as:

$$\mathcal{Y}(t) = (x * w)(t) = \sum_{i=0}^{K-1} w(i) \cdot x_{t-di} \tag{3}$$

where $d$ represents the dilation factor, $K$ indicates the 1D convolutional window size, and $t - di$ denotes the direction of the past with kernel $w : \{0, \ldots, K-1\} \to \mathbb{R}$. However, to construct a deep model with the increased depth of the architecture to capture a more extended history based on the TCN, using skip connections is recommended [7]. Accordingly, the shortcut connections across layers were added to TCNs against the degradation problem, and accuracy saturates as the network converges.

Stacking multiple dilated convolutions enables networks to have extensive receptive fields and capture long-range temporal dependencies with a smaller number of layers [14]. Beside $d_l$ is increased consecutive layers within a block, calculated as $d_l = 2^l$ for layer $l$ in the network. Therefore, each TCN block contains $\gamma$ elements identified based on $R_{field\,max}$, the maximum supported $R_{field}$:

$$\gamma = \left\lceil \log_2 \left( R_{field_{max}} - 1 \right) \right\rceil + 1 \tag{4}$$

As defined in Equation (4), we can consider $\gamma$ a parameter to determine the number of dilations of a TCN block. However, setting the TCN hyperparameters by hand requires an empirical and time-consuming trial-and-error process and is not optimal [7]. In this paper, we reduce this work by automatically searching the hyperparameters. Table 1 lists the hyperparameters automatically searched for in the TCN using Bayesian optimization.

**Table 1.** Search space of the TCN using Bayesian optimization.

| Hyperparameters | Symbol | Choices |
|---|---|---|
| 1D convolutional window size | $K$ | $1, 3, 5, 7, 9$ |
| Number of filters in each convolution layer | $N_i$ | $8, 16, 32, 64, 128, 256, 512$ |
| Number of TCN layers | $N_t$ | $\geq 2$ |
| Dilation factor | $\gamma$ | $\geq 1$ |
| Skip connection | | Yes, No |
| Batch Normalization | | Yes, No |

### 3.3. Stride-TCN

The study focused on energy consumption and was tested on three datasets related to energy consumption. The data analysis found that the energy consumption data are seasonal (Figure 2). If we break down the time series into small components based on that period and compare them, we observe the similarity in the shape of the time series. Therefore, the information in the same positions in the periods has a strong relationship and supports making predictions for the next periods.

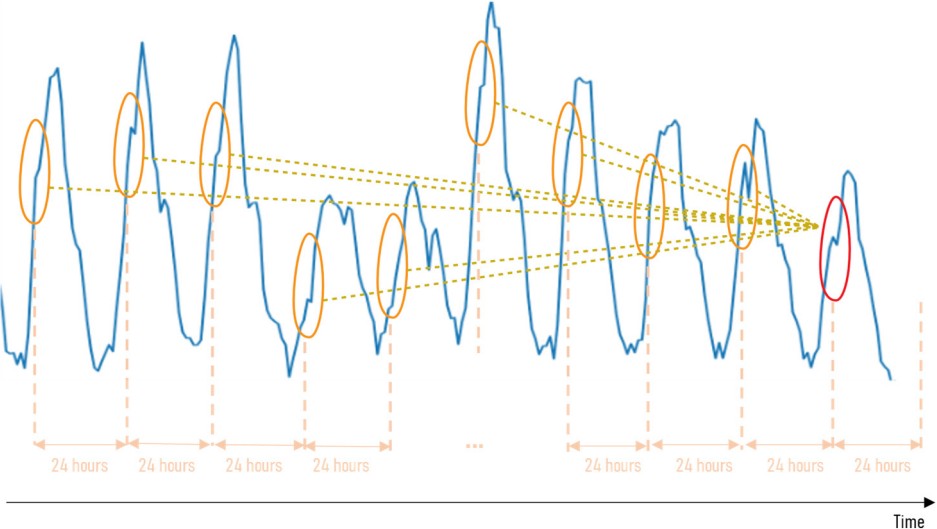

**Figure 2.** Sample in Chonnam National University energy consumption.

The proposed architecture extracts and learns the information at the corresponding locations of each related period, where the model is efficient and reduces the model complexity. Similar to the TCN, the proposed architecture consists of two components: dilated and causal 1D convolutional layers. However, the proposed model was adjusted in the calculation of 1D convolution. The layer $L_n$ is calculated directly based on the lower layer $L_{n-1}$. Node $X_i^{L_n}$ in layer $L_n$ is determined by convolution through a kernel of size $K$ sliding over the layer $L_{n-1}$. Node $X_{i+1}^{L_n}$ in layer $L_n$ is calculated in the same way as $X_i^{L_n}$, but the convolutional position on layer $L_{n-1}$ must be at a distance $S$ from the position of the calculated node $X_i^{L_n}$. This distance is considered the time dependence of the data mentioned above.

The autocorrelation method is used to determine the time dependence $S$ of the data series [15]. We call the model built by this approach heuristic–stride–TCN. Although the model parameters were reduced, the model error was still high compared to the TCN method.

In another approach to determining the time dependence $S$ of the data series, we propose the stride–TCN architecture by applying Bayesian optimization to determine the hyperparameters automatically. We reduced model parameters and errors. Table 2 lists the hyperparameters automatically searched for in the stride-TCN.

**Table 2.** Search space of stride-TCN used by Bayesian optimization.

| Hyperparameters | Symbol | Choices |
| :---: | :---: | :---: |
| Kernal size | $K$ | 1, 3, 5, 7, 9 |
| Number of filters | $N_i$ | 8, 16, 32, 64, 128, 256, 512 |
| Stride | $S$ | $1, 2, 3, \ldots 24$ |
| Dropout rate | $\rho$ | $0, 0.1, \ldots 0.5$ |

An overview of stride–TCN architecture is illustrated in Figure 3.

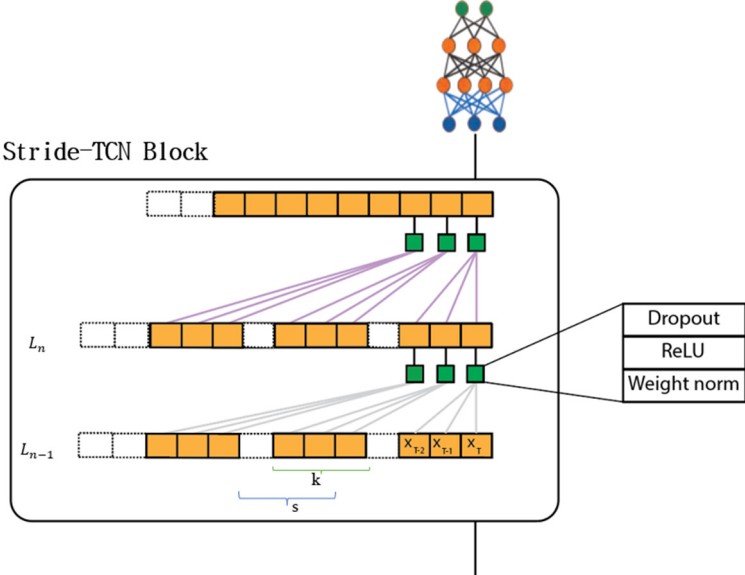

**Figure 3.** Stride-TCN architecture.

### 3.4. Bayesian Optimization

The Bayesian optimization technique probabilistic model $p(\theta \,|\lambda\,)$ of the configuration performs on an evaluation index $\theta$ (i.e., loss or accuracy of the test), given a set of hyperparameters $\lambda$ [16]. Bayesian optimization uses a surrogate model to estimate the function to be optimized, as demonstrated in Algorithm 1.

---

**Algorithm 1** Bayesian Optimization (BO)

---

***Input***: Search space $\Lambda$, black-box function $F$, acquisition function $S$, maximal number of function evaluations $m$

1. $D_0$ = initialize($\Lambda$)
2. **for** n = 1 to $m - |D_0|$ **do**
3.     $p(\theta|\lambda, D)$ = fit predictive model on $D_{n-1}$;
4.     select $x_n$ by optimizing
$\lambda_n = arg \max_{\lambda \in \Lambda} S(\lambda; D_{n-1}, p(\theta|\lambda, D))$;
5. **end for**
6. Query $\theta_n := F(\lambda_n)$;
7. Add observation to data $D_n = D_{n-1} \cup \{\langle \lambda_n, \theta_n \rangle\}$;
8. **return** Best $\lambda^*$

---

Bayesian optimization determines an optimized $\lambda^*$ of the function $F: X \to \mathbb{R}$ to denote a black-box function. Bayesian optimization performs an iterative process to determine the probabilistic $p(\theta|\lambda, D)$ based on the previous observation $D = \{(\lambda_0, \theta_0), (\lambda_1, \theta_1), \ldots, (\lambda_{n-1}, \theta_{n-1})\}$, where it is assumed to only access noisy observations $\theta = F(\lambda) + \varepsilon$ with $\varepsilon \sim \mathcal{N}(0, \sigma_{noise}^2)$. To select the next $\lambda$, the acquisition function–expected improvement $S$ [17] is used to determine a point that maximizes it. Then, it evaluates $F$ at $\lambda_n$, obtains $\theta_n$, updates the probabilistic model, and iterates [18].

### 3.5. Training Procedure

In this paper, the two models TCN and stride-TCN use Bayesian optimization to determine hyperparameters before training the model. The training procedure for each model is described in Algorithm 2, where $W$ denotes model parameters that must be learned, $\lambda^*$ represents the set of optimal adjustable hyperparameters, and loss ($W$) is calculated using the Huber loss.

---

**Algorithm 2** Training procedure

---

*Input*: Search space $\Lambda$, epoch = 100;
1. $\lambda^* = $ **Algorithm 1** ($\Lambda$)
2. Initialize W with $\lambda^*$
3. **for** n = 1 to epoch **do**
4.      Update W based on Huber loss (W)
5. **end for**

---

## 4. Experiments

We evaluated the number of model parameters and predictive power of the stride-TCN compared with the TCN architecture and RNNs, such as LSTMs and GRUs.

### 4.1. Setup

#### 4.1.1. Datasets

We perform experiments on two public and one private dataset for empirical studies. All datasets are available for online access. Because this research focuses on univariate time-series forecasting, we only study time series with a single dimension for each dataset above. Table 3 presents an overview of the corpus statistics.

**Table 3.** Dataset statistics.

| Datasets | Length of Time Series | Total Number of Variables | Attributions |
| --- | --- | --- | --- |
| Dataset 1 | 2,075,259 | 7 | Global active power<br>Global reactive power<br>Voltage<br>Global intensity<br>Submetering 1<br>Submetering 2<br>Submetering 3 |
| Dataset 2 | 8760 | 2 | Energy consumption<br>Outside temperature |
| Dataset 3 | 11,232 | 1 | Energy consumption |

Individual household electric power consumption, Dataset 1 is available online (https://archive.ics.uci.edu/ml/datasets/individual+household+electric+power+consumption, accessed on 17 July 2022). It contains minute-by-minute electric power consumption in one household in France for 47 months (December 2006 to November 2010). The time series includes the total active power consumed, total reactive power consumed, average current intensity, active energy for the kitchen, active energy for the laundry, and active energy for climate control systems [19]. In total, we have 2,075,259 multivariable sequences. For this dataset, 34,589 univariable sequences were used as the study value for the hour–global active power data.

The energy consumption curves of 499 customers from Spain, Dataset 2 is available online (https://fordatis.fraunhofer.de/handle/fordatis/215, accessed on 17 July 2022). The dataset contains hourly energy consumption data, outside temperatures for the region, and the metadata for 499 customers in Spain for about one year (1 January 1 to 31 December 2019). The entire dataset consists of 8760 data points. The energy consumption data was used for this dataset as the study value.

The CNU energy consumption, Dataset 3 is available online (https://github.com/andrewlee1807/tcns-with-nas/tree/main/Dataset/cnu-dataset, accessed on 5 August 2022). It contains a real-world dataset with energy consumption values of 90 locations at CNU, collected continuously hourly for 1.3 years (from 1 January 2021 to 14 January 2022). Each location has information for 11,232 data points. In this research, we focus on the total electricity consumption of a particular location: Engineering-Building-07.monitor_02.

For each dataset, it is necessary to conduct preprocessing procedures before the training process because the power values in the datasets are relatively high; for example, in Dataset 3, *mean* = 130.48 and *std* = 46.97. Therefore, to avoid overflow, increased computational cost, and dataset distortion, each dataset refers to the rescaling of the features to a range of [0, 1] using the min–max normalization, calculated as follows (5):

$$z_i = \frac{x_i - \min(x)}{\max(x) - \min(x)} \tag{5}$$

where $x = (x_1, \ldots, x_n)$ and $z_i$ is the $i^{th}$ normalized data point.

We also evaluated the time series forecasting task on three datasets in this experiment. More specifically, most models choose an input length of 168 h and output length of 1 to 84 h. Each dataset was split into a training set (80%), validation set (10%), and testing set (10%) in chronological order.

### 4.1.2. Model variants

We conducted experiments on two model variants, the heuristic–stride–TCN and stride-TCN. Depending on each dataset, the models have the appropriate configuration. The heuristic–stride–TCN is built in two hidden layers relying on a pattern of individual data to determine the value of the stride. Table 4 presents the configuration for the three datasets.

**Table 4.** Configuration for the heuristic-stride-TCN for three datasets.

| Hyperparameters | Dataset 1 | Dataset 2 | Dataset 3 |
|---|---|---|---|
| 1D convolutional window size | 3 | 3 | 3 |
| Number of filters in each convolution layer | 32 | 32 | 32 |
| Stride 1 | 12 | 24 | 24 |
| Stride 2 | 7 | 7 | 7 |

For each dataset, we experimented on three different stride–TCN models, with two hidden layers, three hidden layers, and four hidden layers. The stride–TCN was built relying on Bayesian optimization to determine the optimal hyperparameters in Table 2.

### 4.1.3. Evaluation Metrics

Two evaluation metrics, the *MAE* and *MSE* for univariate forecasting, are employed, defined as follows:

$$MAE = \frac{1}{n} \sum_{i=1}^{n} \left| Y_i^{real} - Y_i^{predict} \right| \tag{6}$$

$$MSE = \frac{1}{n} \sum_{i=1}^{n} \left( Y_i^{real} - Y_i^{predict} \right)^2 \tag{7}$$

The implementations of the proposed methods were built based on the Keras library with a Tensorflow backend. All models were trained and tested on four Nvidia Quadro RTX A5000 24 GB GPUs. The source code is available online (https://github.com/andrewlee1807/tcns-with-nas).

### 4.2. Experimental Results and Comparison
#### 4.2.1. Baselines and Configurations

The LSTM, GRU, and TCN models are included for an evaluation to build a baseline test benchmark. The LSTM model is built with two hidden LSTM layers. The first LSTM layer identifies 200 hidden nodes, and the second LSTM layer identifies 150 nodes. The final layer is dense. The GRU model is built with two hidden GRU layers. The first GRU layer identifies 103 hidden nodes, and the second GRU layer identifies 103 nodes. The

final layer is dense with the rectified linear unit activation function. In both LSTM and GRU models, the dropout is 25%, the optimizer is Adam, and the loss function is MSE to train the model. The TCN model was built using Bayesian optimization to determine the optimal hyperparameters in Table 1.

Both the TCN baseline model and our proposed stride-TCN are built with BO for the optimal hyperparameters. To find an optimum robust model that is general to an arbitrary training process, we keep the hyperparameters on the training phase unchanged and only search for TCN architecture involving kernel size, dilation, and the number of layers which are the most important hyperparameters. On the other hand, we further shrink the search space of the Stride-TCN family where only stride, the number of layers, kernel size, and whether using dropout is considered. Our purpose is to drive the search process to focus more on finding the best stride hyperparameters yet not too much to suppress the contribution of other important hyperparameters. The range for each hyperparameter of the stride-TCN family is given in Table 2. Additional TCN and stride-TCN use Huber loss during training, parameter $\delta$ is set to 1 for all cases.

As mentioned above, configurations of the training phase are kept unchanged for all learning models. For details, we set the starting learning rate at 0.001 and reduce it by 1% when there is no improvement in validation loss. We train the model for 100 epochs with the Adam optimizer batch size of 32. In addition, we apply early stopping when there is no improvement after 20 epochs.

### 4.2.2. Results and Analysis

The comparison was made using the MSE and MAE for forecast horizons set from the first hour to the 84th hour. The forecast horizon is the length of time into the future for which forecasts are to be prepared. To avoid abundant observations, we hence only report a result at particular time steps, which are 1, 12, 24, 36, 48, 60, 72, and 84 h. In time series forecasting, larger horizons make forecast prediction more challenging. Thus, the experiments offer a detailed analysis of the results in this vast horizon. We compared the results of the proposed method with other algorithms (LSTM, GRU, and TCN) to prove the effectiveness of this approach. The best results (lower values are better) per method are highlighted in red.

Table 5 reports the MSE and MAE values of predicted energy consumption on the CNU dataset (dataset 1) over time steps from baselines LSTM, GRN, TCN, and our proposed stride–TCNs. As clearly depicted in Table 5, our stride–TCNs steadily achieve the lowest errors between 60 h and 84 h. Notably, the stride–TCN with two layers model reaches the lowest prediction error of 84 h. Compared to the LSTM, GRU, heuristic–stride–TCN, and TCN baseline, our auto–stride–2 layers are 2.07%, 7.19%, 30.05%, and 32.7% better in terms of average relative errors, respectively. In the case of dataset 2, the TCN baseline has demonstrated its superior when substantially better than other models for short-time prediction, as in Table 6 However, the limitations become clear when experimenting on dataset 1, when our proposed architecture could not overcome baseline models, as shown in Table 7. This indicates the lack of complexity of the model to capture information well in a large dataset, as well known as the underfitting phenomenon. Overall, our proposed architecture achieved on-par results with other baseline models on time series forecasting, considering the error is slightly higher or even lower than the baselines' error in some specific settings.

We obtain a significant reduction in model complexity with stride–TCNs, as seen in Table 4. Remarkably, our heuristic–stride–TCN for long-time forecasting (84 h) has approximately 6K parameters, which is only 1.6%, 5%, and 1% compared to the number of parameters from the baseline LSTM, GRU, and TCN, respectively. We also note that the model's complexity depends on the length of both history and forecast horizontal, so each model's complexity on different datasets is variant. Details are given in Table 8 (dataset 2), Table 9 (dataset 3) and Table 10 (dataset 1).

**Table 5.** Performance (MSE and MAE) of all models on the CNU dataset.

| Method | Time Prediction | | | | | | | | | | | | | | | |
|---|---|---|---|---|---|---|---|---|---|---|---|---|---|---|---|---|
| | 1 h | | 12 h | | 24 h | | 36 h | | 48 h | | 60 h | | 72 h | | 84 h | |
| | MSE | MAE | MSE | MAE | MSE | MAE | MSE | MAE | MSE | MAE | MSE | MAE | MSE | MAE | MSE | MAE |
| LSTM | 0.0020 | 0.0297 | 0.0109 | 0.0687 | 0.0097 | 0.0670 | 0.0113 | 0.0741 | 0.0140 | 0.0813 | 0.014 | 0.083 | 0.0140 | 0.0830 | 0.0145 | 0.0861 |
| GRU | 0.0022 | 0.0309 | 0.0109 | 0.0702 | 0.0128 | 0.0764 | 0.0145 | 0.0818 | 0.0157 | 0.0853 | 0.0156 | 0.0866 | 0.0156 | 0.0866 | 0.0153 | 0.0871 |
| TCN | 0.0020 | 0.0298 | 0.0113 | 0.0718 | 0.0084 | 0.0639 | 0.0107 | 0.0714 | 0.0157 | 0.0852 | 0.0168 | 0.0886 | 0.0179 | 0.0927 | 0.0203 | 0.0980 |
| Heuristic–stride–TCN | 0.0121 | 0.0799 | 0.0191 | 0.1012 | 0.0208 | 0.1049 | 0.0207 | 0.1044 | 0.0212 | 0.105 | 0.0213 | 0.1053 | 0.0213 | 0.1053 | 0.0211 | 0.1047 |
| Stride–TCN | | | | | | | | | | | | | | | | |
| 2 layers | 0.0025 | 0.034 | 0.0115 | 0.0763 | 0.0136 | 0.0795 | 0.0129 | 0.0793 | 0.0148 | 0.0843 | 0.0137 | 0.0823 | 0.0133 | 0.0793 | 0.0142 | 0.0849 |
| 3 layers | 0.0024 | 0.0331 | 0.012 | 0.0725 | 0.0109 | 0.071 | 0.0149 | 0.0837 | 0.0153 | 0.0841 | 0.0165 | 0.0865 | 0.0123 | 0.0771 | 0.016 | 0.0868 |
| 4 layers | 0.0023 | 0.0322 | 0.0117 | 0.0765 | 0.0107 | 0.0727 | 0.0139 | 0.0833 | 0.0154 | 0.0828 | 0.0174 | 0.0922 | 0.0169 | 0.0888 | 0.016 | 0.0865 |

**Table 6.** Performance (MSE and MAE) of all models on the Spain dataset.

| Method | Time Prediction | | | | | | | | | | | | | | | |
|---|---|---|---|---|---|---|---|---|---|---|---|---|---|---|---|---|
| | 1 h | | 12 h | | 24 h | | 36 h | | 48 h | | 60 h | | 72 h | | 84 h | |
| | MSE | MAE | MSE | MAE | MSE | MAE | MSE | MAE | MSE | MAE | MSE | MAE | MSE | MAE | MSE | MAE |
| LSTM | 0.0081 | 0.0677 | 0.0169 | 0.0971 | 0.0165 | 0.096 | 0.0169 | 0.0974 | 0.016 | 0.0949 | 0.0173 | 0.0982 | 0.0158 | 0.0949 | 0.0163 | 0.0957 |
| GRU | 0.0095 | 0.0723 | 0.0161 | 0.0937 | 0.0167 | 0.0955 | 0.0188 | 0.1015 | 0.0197 | 0.1037 | 0.02 | 0.1044 | 0.0198 | 0.104 | 0.0205 | 0.1061 |
| TCN | 0.008 | 0.0672 | 0.0149 | 0.0894 | 0.0156 | 0.0921 | 0.0166 | 0.096 | 0.0173 | 0.0981 | 0.0187 | 0.1015 | 0.0182 | 0.1 | 0.0182 | 0.1005 |
| Heuristic–stride–TCN | 0.0247 | 0.1166 | 0.0403 | 0.1562 | 0.0424 | 0.1607 | 0.0421 | 0.1604 | 0.0426 | 0.1615 | 0.0426 | 0.1613 | 0.0428 | 0.1621 | 0.0432 | 0.1626 |
| Stride-TCN | | | | | | | | | | | | | | | | |
| 2 layers | 0.0116 | 0.0792 | 0.0175 | 0.0989 | 0.0209 | 0.1071 | 0.0192 | 0.1036 | 0.0215 | 0.1083 | 0.0199 | 0.106 | 0.02 | 0.1046 | 0.0202 | 0.107 |
| 3 layers | 0.0179 | 0.0989 | 0.018 | 0.1004 | 0.0223 | 0.1104 | 0.0181 | 0.1006 | 0.0202 | 0.1079 | 0.0203 | 0.1047 | 0.0203 | 0.1047 | 0.0213 | 0.109 |
| 4 layers | 0.0085 | 0.0717 | 0.0178 | 0.0998 | 0.0205 | 0.1089 | 0.0199 | 0.1043 | 0.0204 | 0.1102 | 0.02 | 0.1049 | 0.0202 | 0.1064 | 0.0184 | 0.1001 |

**Table 7.** Performance (MSE and MAE) of all models on the Household dataset.

| Method | Time Prediction | | | | | | | | | | | | | | | |
|---|---|---|---|---|---|---|---|---|---|---|---|---|---|---|---|---|
| | **1 h** | | **12 h** | | **24 h** | | **36 h** | | **48 h** | | **60 h** | | **72 h** | | **84 h** | |
| | **MSE** | **MAE** | **MSE** | **MAE** | **MSE** | **MAE** | **MSE** | **MAE** | **MSE** | **MAE** | **MSE** | **MAE** | **MSE** | **MAE** | **MSE** | **MAE** |
| LSTM | 0.0056 | 0.0517 | 0.0081 | 0.0656 | 0.0083 | 0.066 | 0.0085 | 0.0673 | 0.0086 | 0.0679 | 0.0091 | 0.0714 | 0.0089 | 0.0694 | 0.009 | 0.0711 |
| GRU | 0.0065 | 0.0569 | 0.0079 | 0.0657 | 0.0082 | 0.068 | 0.0083 | 0.0687 | 0.0085 | 0.0693 | 0.0085 | 0.0696 | 0.0086 | 0.0704 | 0.009 | 0.0732 |
| TCN | 0.0052 | 0.0496 | 0.0082 | 0.0658 | 0.0085 | 0.0671 | 0.0087 | 0.0689 | 0.0089 | 0.0693 | 0.0095 | 0.0724 | 0.0091 | 0.0712 | 0.009 | 0.0715 |
| Heuristic–stride–TCN | 0.0104 | 0.0798 | 0.0115 | 0.0863 | 0.0116 | 0.0871 | 0.0116 | 0.0873 | 0.0116 | 0.0874 | 0.0116 | 0.087 | 0.0116 | 0.0871 | 0.0117 | 0.0874 |
| Stride-TCN | | | | | | | | | | | | | | | | |
| 2 layers | 0.0063 | 0.0598 | 0.0088 | 0.0687 | 0.0089 | 0.0725 | 0.0088 | 0.0707 | 0.009 | 0.0727 | 0.0093 | 0.0741 | 0.0093 | 0.0726 | 0.0093 | 0.0743 |
| 3 layers | 0.0055 | 0.0503 | 0.0086 | 0.0707 | 0.0088 | 0.072 | 0.009 | 0.0736 | 0.009 | 0.0726 | 0.009 | 0.0732 | 0.0093 | 0.0757 | 0.0096 | 0.078 |
| 4 layers | 0.0054 | 0.0508 | 0.0101 | 0.0808 | 0.0104 | 0.0789 | 0.0094 | 0.073 | 0.0096 | 0.0751 | 0.0093 | 0.0754 | 0.0101 | 0.0808 | 0.0097 | 0.0777 |

**Table 8.** Models' complexity in the Household dataset (dataset 1), demonstrated by the number of parameters.

| Model | Time Prediction | | | | | | | |
|---|---|---|---|---|---|---|---|---|
| | **1 h** | **12 h** | **24 h** | **36 h** | **48 h** | **60 h** | **72 h** | **84 h** |
| LSTM | 372,351 | 374,012 | 375,824 | 377,636 | 379,448 | 381,260 | 383,072 | 384,884 |
| GRU | 103,747 | 104,462 | 105,242 | 106,022 | 106,802 | 107,582 | 108,362 | 109,142 |
| TCN | 23,681 | 1,495,436 | 269,144 | 646,052 | 647,600 | 1,501,628 | 650,696 | 11,716 |
| Heuristic–stride–TCN | 3521 | 3884 | 4280 | 4676 | 5072 | 5468 | 5864 | 6260 |
| Stride-TCN | | | | | | | | |
| 2 layers | 2081 | 30,540 | 31,320 | 32,100 | 32,880 | 9692 | 34,440 | 3492 |
| 3 layers | 58,817 | 59,532 | 60,312 | 61,092 | 61,872 | 62,652 | 63,432 | 64,212 |
| 4 layers | 87,809 | 1268 | 89,304 | 23,556 | 1992 | 91,644 | 1808 | 2316 |

**Table 9.** Models' complexity in CNU dataset (dataset 3), demonstrated by the number of parameters.

| Model | Time Prediction | | | | | | | |
|---|---|---|---|---|---|---|---|---|
| | **1 h** | **12 h** | **24 h** | **36 h** | **48 h** | **60 h** | **72 h** | **84 h** |
| LSTM | 372,351 | 374,012 | 375,824 | 377,636 | 379,448 | 381,260 | 383,072 | 384,884 |
| GRU | 103,747 | 104,462 | 105,242 | 106,022 | 106,802 | 107,582 | 108,362 | 109,142 |
| TCN | 6433 | 88,652 | 1,037,720 | 580,004 | 353,968 | 355,516 | 587,720 | 589,268 |
| Heuristic–stride–TCN | 3521 | 3884 | 4280 | 4676 | 5072 | 5468 | 5864 | 6260 |
| Stride-TCN | | | | | | | | |
| 2 layers | 593 | 8108 | 800 | 32,100 | 1016 | 33,660 | 34,440 | 10,484 |
| 3 layers | 1081 | 59,532 | 60,312 | 61,092 | 16,624 | 62,652 | 63,432 | 64,212 |
| 4 layers | 38,401 | 88,524 | 89,304 | 90,084 | 90,864 | 91,644 | 2208 | 68,500 |

**Table 10.** Models' complexity in Spain dataset (dataset 2), demonstrated by the number of parameters.

| Model | Time Prediction | | | | | | | |
|---|---|---|---|---|---|---|---|---|
| | **1 h** | **12 h** | **24 h** | **36 h** | **48 h** | **60 h** | **72 h** | **84 h** |
| LSTM | 372,351 | 374,012 | 375,824 | 377,636 | 379,448 | 381,260 | 383,072 | 384,884 |
| GRU | 103,747 | 104,462 | 105,242 | 106,022 | 106,802 | 107,582 | 108,362 | 109,142 |
| TCN | 345,857 | 374,988 | 260,824 | 319,076 | 188,528 | 52,700 | 814,280 | 1,504,724 |
| Heuristic–stride–TCN | 3521 | 3884 | 4280 | 4676 | 5072 | 5468 | 5864 | 6260 |
| Stride-TCN | | | | | | | | |
| 2 layers | 7745 | 692 | 31,320 | 2676 | 9296 | 3084 | 34,440 | 10,484 |
| 3 layers | 58,817 | 5,9532 | 60,312 | 4548 | 1504 | 62,652 | 1720 | 17,812 |
| 4 layers | 22,401 | 6012 | 89,304 | 90,084 | 90,864 | 91,644 | 92,424 | 93,204 |



Figure 4 illustrates the correlation between performance and complexity between seven models on the CNU dataset, including the LSTM, GRU, and TCN baseline, together with our proposed stride–TCN and heuristic–stride–TCN. In most cases, the baseline TCN has the largest number of parameters, followed by the LSTM and GRU. On the contrary, the heuristic–stride–TCN model constantly reaches the smallest number of parameters. Besides, the family of the stride–TCN also has a relatively small number of parameters compared to baseline models. Figure 4 also demonstrates that using BO usually leads to a model with better performance but suffers the model's complexity trade-off. Another statement that can be drawn here is that the baseline models outperform stride-TCN only in the task of short-term forecasting since the difference in MSE between baseline and the proposed architecture is not trivial later. Generally, the results confirm that our proposed TCN architecture yields a family of lightweight models capable of being implemented on constrained resource devices. Last but not least, among models in the stride–TCN family, the heuristic–stride–TCN does favor not only less complexity but also the fastest in terms of training because the stride hyperparameter can be predefined by data's pattern instead of being searched by BO.

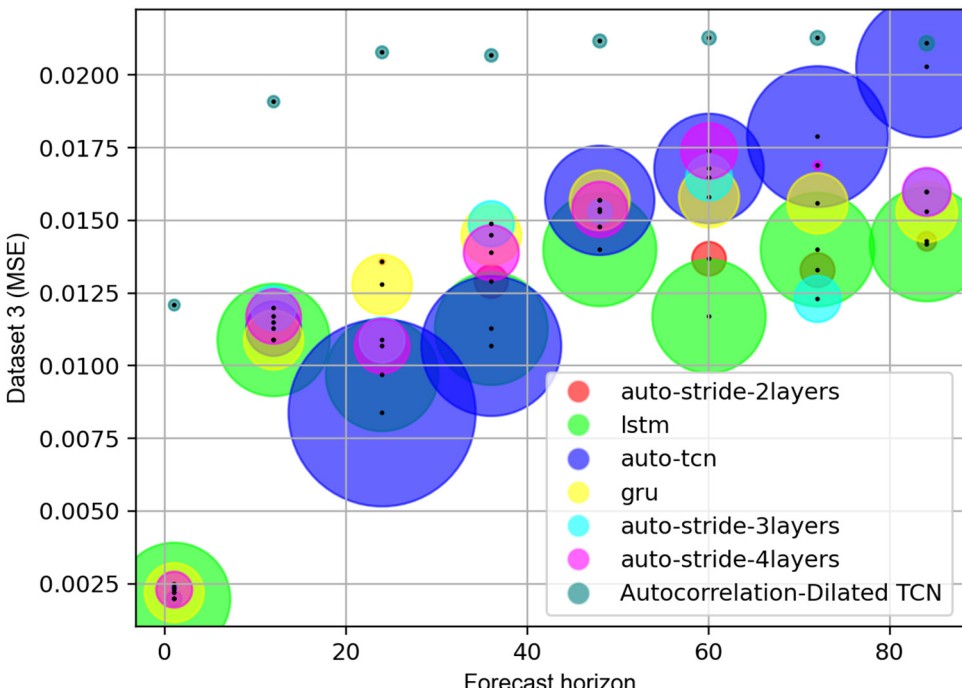

**Figure 4.** Comparison between models with respect to performance and complexity. The comparison is conducted on the CNU dataset. The x-axis depicts MSE error, and the y-axis depicts the forecast horizontal at eight milestones (1, 12, 24, 36, 48, 60, 72, and 84 h). A circle represents a model whose color represents its category, and the circle radius describes its complexity.

## 5. Conclusions

This paper presents three contributions. First, we propose a lightweight TCN family with the stride mechanism; secondly, we introduce a new dataset about electrical energy consumption along with its benchmark; and thirdly, we search for a robust model based on Bayesian Optimization. The experiments have shown that our architecture achieves comparable results on small and medium datasets while significantly reducing model complexity compared to baselines. We argue that the performance on the large dataset is not high as expected because we limit the number of dilation layers which makes our model underfit. Importantly, our results provide evidence for the hypothesis that highly correlated time points are crucial for the forecasting task. Furthermore, we suggest that the stride factors should be trained alongside the model's parameters to make it adaptable to various datasets. This assumption might be addressed in future studies.

**Author Contributions:** Conceptualization, L.H.A. and J.Y.K.; methodology, L.H.A., G.H.Y. and D.T.V.; software, L.H.A.; validation, J.Y.K., G.H.Y. and J.C.Y.; formal analysis, D.T.V.; investigation, J.I.L. and J.C.Y.; resources, J.S.K.; data curation, L.H.A. and J.Y.K.; writing—original draft preparation, L.H.A.; writing—review and editing, J.Y.K., J.S.K. and J.C.Y.; visualization, L.H.A.; supervision, J.Y.K. and J.S.K.; project administration, J.C.Y., J.S.K. and J.Y.K.; funding acquisition, J.I.L., J.S.K. and J.Y.K. All authors have read and agreed to the published version of the manuscript.

**Funding:** This work was supported by the Korea Electric Power Research Institute (KEPRI) grant funded by the Korea Electric Power Corporation (KEPCO) (No. R20IA02). And this work was supported by the Institute of Information & communications Technology Planning & Evaluation (IITP) grant funded by the Korean government (MSIT) (No. 2021-0-02068, Artificial Intelligence Innovation Hub).

**Institutional Review Board Statement:** Not applicable.

**Informed Consent Statement:** Not applicable.

**Data Availability Statement:** Individual household electric power consumption is available online at https://archive.ics.uci.edu/ml/datasets/individual+household+electric+power+consumption (accessed on 17 July 2022). The energy consumption curves of 499 customers from Spain are available online at https://fordatis.fraunhofer.de/handle/fordatis/215 (accessed on 17 July 2022). The CNU energy consumption is available online at https://github.com/andrewlee1807/tcns-with-nas/tree/main/Dataset/cnu-dataset (accessed on 5 August 2022).

**Conflicts of Interest:** The authors declare no conflict of interest.

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
