# Peer review of "Stride-TCN for Energy Consumption Forecasting and Its Optimization"

_applsci, doi:10.3390/app12199422_

Round 1
Reviewer 1 Report
To Authors
This study proposes a stride-dilation mechanism for TCN that favors a lightweight model yet still achieves on-pair accuracy with the heavy counterparts. The topic is relevant to this field and complements previous research with a more efficient model. The conclusions are consistent with the arguments and results presented in the paper and give a good presentation of the topic. References are appropriate.
The paper is well written, but minor corrections are needed:
The abstract needs to be corrected;
Typos must be corrected (line 201 etc.);
In Figure 2, improve the contrast;
Study and pick up the latest references.
Author Response
- The abstract needs to be corrected;
The authors have revised the abstract and clarified each statement. We have ensured that the abbreviate quantitative results shown in the abstract are the key results obtained from the experiment. As standard, the abstract has briefly described the research's topic, scope, purpose and main results.
- Typos must be corrected (line 201 etc.);
All typo errors have been fixed.
- In Figure 2, improve the contrast
We have improved the contrast of figure 2 as requested.
- Study and pick up the latest references.
We have added recent research references.

Reviewer 2 Report
What is the difference between Stride TCN and a Convolutional network with different convolutional layers having a stride?
Page 5 of 15, paragraph 1 states that “The data analysis found that the energy consumption data are cyclical”. Was this data analysis only through manual observation OR some other statistical data analysis techniques were applied to find out the cyclical pattern?
Reference 4 seems incomplete.
Some typos can be found, which should be improved, e.g.:
Page 1 of 15: line 28
Page 6 of 15: line 201 and page 7 of 15: line 225 (some non-English characters)
Author Response
- What is the difference between Stride TCN and a Convolutional network with different convolutional layers having a stride?
Figure 1
To answer the question, we first suppose that the "Convolutional network with different convolutional layers having a stride" that the reviewer mentioned here is the "Standard Convolutional network", not use the "Casual Convolutional layer". Our Stride-TCN (Stride Temporal convolutional networks), where Stride is a parameter of the neural network's filter that controls how the filter convolves around the input volume, is inspired from the conventional TCN and acquired it the stride mechanism, explained in lines 84-89.
Although "Stride TCN" and "CNN with different convolutional layers having a stride" both involve convolutional operators, Stride TCN has a remarkable point. As depicted in Figure 1, Stride-TCN uses causal convolutions with individual strides parameters at different layers. Causal convolutions are a type of convolution used for temporal data, ensuring that the output at time t derives only from inputs from time t – 1, while standard convolution does not take the direction of convolution into account. Stride-TCN ensures the output value must only depend on values positioned earlier in the input sequence to prevent information leakage from future to past.
- Page 5 of 15, paragraph 1 states that "The data analysis found that the energy consumption data are cyclical". Was this data analysis only through manual observation OR some other statistical data analysis techniques were applied to find out the cyclical pattern?
Figure 2
After revising the manuscript and investigating other studies in time-series data, we supposed that the term "cyclical" is incorrect and had it replaced by the term "seasonal", at line .
In particular, We have analyzed data using the time-series decomposition technique, calculating correlation and manually observing the seasonal pattern by thresholding the correlation values. We have also inspected several public datasets about energy consumption and found that most of them possess seasonal patterns.
For example, the Figure above shows a sample in CNU energy consumption dataset, which is decomposed into three components, trend, seasonal and residual. The above result shows that the seasonal component's value is 24 hours.
- Reference 4 seems incomplete.
We have revised the information.
- Some typos can be found, which should be improved, e.g.: Page 1 of 15: line 28; Page 6 of 15: line 201 and page 7 of 15: line 225 (some non-English characters)
All typo errors have been fixed.

Reviewer 3 Report
This paper proposes a novel model that greatly reduces the number of involved parameters. The results indicate that the proposed stride TCN architecture has performance comparable to state-of-the-art methods. Another contribution is an introduction of a new dataset which can be a fine addition that can be quite useful to the research community. Overall, the paper is well-written and easy to understand. It’s an easy decision, but some minor recommendations should be addressed before publication.
1. Line 28: There seems to be a typo at the end.
2. Line 33: It's better to rephrase the sentence “… such as machine learning and deep learning”. To me, the terminology is not accurate as deep learning is a part of machine learning. Please also consider other places in the main text such as line 76, etc.
3. Line 38: “such as recurrent neural networks (RNNs) and convolutional neural 37 networks (CNNs)”. Usually, CNN includes a feature extractor pipeline and mainly this (alone) isn’t a very good approach for time series forecasting. It is suggested to rephrase this line and add methods specifically related to time-series forecasting such as LSTM, GRU, Transformers, BERT, etc.
4. Line 60: “This paper is constructed as follows.” The following paragraph is missing information of section 1. It is suggested either to add details for section 1 or rephrase the previously quoted sentence to “The rest of the paper is organized as follows” etc.
5. It is suggested to follow the same norm for all the web links. It's better to use a footnote for all the web links. For example, Ref. [4] is a weblink, and the footer [1] is also a web link. In some places, the web links are given in the main text i.e., line 230, line 238, line 244.
6. Line 96: “robust in terms of model complexity”. This part of the sentence is confusing, it's better to write “lower complexity” and “robust performance”.
7. Line 105: To achieve consistency and avoid confusion to readers, it is suggested to also add round brackets with ? = ?0Ì‚, … ?Ì‚P. Moreover, add a comma before ?Ì‚P.
8. Equation 1: Seems to be mathematically inaccurate as hat is missing in ?0 , . . . , ?_p . Either remove this as it seems to be redundant as function mapping is already provided. Or it is suggested change it to simply Y=f(X).
9. Line 110: “Building function ? is the process of learning to find a network ? from a set of time 110 series”. This seems to be erroneous, as the learning process doesn’t find a network rather the learning process learns the parameters as mentioned in line 88.
10. Line 111: the time series X declared doesn’t align with X represented in line 105. The X in line 111 seems to be the form of a Set. It is suggested to use common notation throughout the manuscript i.e., you may also covert X in line 105 to a set notation (use curly brackets).
11. Line 114: Moreover, add a comma before ?̂P.
12. There is a miss-aligned “:” on line 122.
13. In Equation 2: I am a bit confused here, what is “y” is this the same “Y” mentioned in line 105? Moreover, the loss function notation i.e., L_delta must be aligned as mentioned in line 114.
14. In Equation 3: Please avoid using a dot product notation i.e., “∙”, as it may confuse the readers. It's better to use multiplication or no symbol at all. Moreover, what are x and f? Additionally, it's better to change the notation of f as it has already been used to represent a function.
15. Line 149: The notation d doesn’t align with Figure 1. Please make it consistent throughout the manuscript.
16. Figure 1: What is the end range/value of L?
17. Please use a consistent representation of K in Equation 3, Table 1, Table 2, Line 140, etc.
18. Figure 2: It would be more convenient if the time scale in terms of weeks, months, or years is mentioned. It will provide a better picture to readers.
19. Line 174: “The ?? layers are calculated directly based”. This sentence is not clear. Do you mean the layer calculation or layer parameters?
20. Algorithm 1, Algorithm 2: Some notations should be explained as the notations are not standard i.e., both arrows, := . Moreover, try to use consistent loop notation for both algorithms.
21. Line 201, 225, 248: There seems to be a problem with the reference.
22. Algorithm 2, Line 4: Please mention the loss notation as described earlier in the manuscript.
23. Equations 6 and 7: These Equations are conflicting with ?0 , . . . , ?p. It creates confusion about whether y_i in these equations is an element or a vector.
24. In line 287: If the loss function for training is MSE, then why Huber loss is introduced on line 121?
25. Line 318: The mentioned Figure 3 doesn’t indicate the error rates. It seems that the figure reference is inaccurate. Please reconfirm all the figure references.
26. Please remove the line “In summary” from the conclusion.
Author Response
- Line 28: There seems to be a typo at the end.
This typo has been fixed at line 28.
- Line 33: It's better to rephrase the sentence "… such as machine learning and deep learning". To me, the terminology is not accurate as deep learning is a part of machine learning. Please also consider other places in the main text such as line 76, etc.
These typos have been fixed at lines 36, 75 and 76.
- Line 38: "such as recurrent neural networks (RNNs) and convolutional neural 37 networks (CNNs)". Usually, CNN includes a feature extractor pipeline and mainly this (alone) isn't a very good approach for time series forecasting. It is suggested to rephrase this line and add methods specifically related to time-series forecasting such as LSTM, GRU, Transformers, BERT, etc.
This typo has been fixed at line 36.
- Line 60: "This paper is constructed as follows." The following paragraph is missing information of section 1. It is suggested either to add details for section 1 or rephrase the previously quoted sentence to "The rest of the paper is organized as follows" etc.
This typo has been fixed at line 59.
- It is suggested to follow the same norm for all the web links. It's better to use a footnote for all the web links. For example, Ref. [4] is a weblink, and the footer [1] is also a web link. In some places, the web links are given in the main text i.e., line 230, line 238, line 244.
Footnote has been used to cite all the web links at lines 29, 239, 247, 251 and 284.
- Line 96: "robust in terms of model complexity". This part of the sentence is confusing, it's better to write "lower complexity" and "robust performance".
This typo has been fixed at line 95-96.
- Line 105: To achieve consistency and avoid confusion to readers, it is suggested to also add round brackets with ?= ?0Ì‚, … ?Ì‚ Moreover, add a comma before ?Ì‚P.
This equation has been fixed at line 116.
- Equation 1: Seems to be mathematically inaccurate as hat is missing in ?0 , . . . , ?_p . Either remove this as it seems to be redundant as function mapping is already provided. Or it is suggested change it to simply Y=f(X).
The equation (1) has been fixed.
- Line 110: "Building function ?is the process of learning to find a network ? from a set of time 110 series". This seems to be erroneous, as the learning process doesn't find a network rather the learning process learns the parameters as mentioned in line 88.
This error has been fixed at line 121.
- Line 111: the time series X declared doesn't align with X represented in line 105. The X in line 111 seems to be the form of a Set. It is suggested to use common notation throughout the manuscript i.e., you may also covert X in line 105 to a set notation (use curly brackets).
This error has been fixed at line 114.
- Line 114: Moreover, add a comma before ?Ì‚
This error has been fixed at line 123.
- There is a miss-aligned ":" on line 122.
This error has been fixed at line 132.
- In Equation 2: I am a bit confused here, what is "y" is this the same "Y" mentioned in line 105? Moreover, the loss function notation i.e., L_delta must be aligned as mentioned in line 114.
in equation 2 and mentioned in line 116 are the same, the equation (2) has been fixed by using uppercase Y.
- In Equation 3: Please avoid using a dot product notation i.e., "∙", as it may confuse the readers. It's better to use multiplication or no symbol at all. Moreover, what are x and f? Additionally, it's better to change the notation of f as it has already been used to represent a function.
We have changed the notation to to avoid ambiguity. Therefore, is a sequence input and is a filter. The change has been updated from lines 148 to 152 and modified Equation 3.
- Line 149: The notation d doesn't align with Figure 1. Please make it consistent throughout the manuscript.
has been used consistently in Figure 1 and throughout the manuscript, lines 161-162.
- Figure 1: What is the end range/value of L?
As explained in line 161, in TCN, where is the number of layers.
- Please use a consistent representation of K in Equation 3, Table 1, Table 2, Line 140, etc.
We have revised and used a consistent representation of in equation (3), Table 1, 2 and Line 152.
- Figure 2: It would be more convenient if the time scale in terms of weeks, months, or years is mentioned. It will provide a better picture to readers.
We have edited Figure 2 to provide better intuition to readers.
- Line 174: "The ??layers are calculated directly based". This sentence is not clear. Do you mean the layer calculation or layer parameters?
Figure 3
is layer used in calculation. We have revised this information in lines 186-187.
- Algorithm 1, Algorithm 2: Some notations should be explained as the notations are not standard, i.e., both arrows, := . Moreover, try to use consistent loop notation for both algorithms.
We have edited Algorithms 1 and 2.
- Line 201, 225, 248: There seems to be a problem with the reference.
We have corrected the references at lines 213, 236 and 257.
- Algorithm 2, Line 4: Please mention the loss notation as described earlier in the manuscript.
We have mentioned the loss notation as described earlier in Algorithm 2.
- Equations 6 and 7: These Equations are conflicting with ?0 , . . . , ? It creates confusion about whether y_i in these equations is an element or a vector.
We have capitalized the notation to avoid ambiguity, and in equations 6 and 7 is an element.
- In line 287: If the loss function for training is MSE, then why Huber loss is introduced on line 121?
In this study, we have employed Huber loss as an objective function to train TCN models, while MSE is for training LSTMs and GRUs. The decision to use different objective functions here is based on our experimental results and prior research evidence. For fair comparisons, However, we only use MSE as a metric to evaluate the models. The explanation has been added to lines 282 and 294.
- Line 318: The mentioned Figure 3 doesn't indicate the error rates. It seems that the figure reference is inaccurate. Please reconfirm all the figure references.
We have revised and fixed the ‘Figure 3’ to ‘Table 5’ references at line 326.
- Please remove the line "In summary" from the conclusion.
We have removed this line, the change can be checked at line 387.
